# Preclinical Advances in LNP-CRISPR Therapeutics for Solid Tumor Treatment

**DOI:** 10.3390/cells13070568

**Published:** 2024-03-24

**Authors:** Shuting Wang, Yuxi Zhu, Shi Du, Yunsi Zheng

**Affiliations:** 1School of Pharmacy, Hainan Medical University, Haikou 571199, China; hy0207113@hainmc.edu.cn; 2Department of Pediatrics, University Hospitals Rainbow Babies & Children’s Hospital, Cleveland, OH 44106, USA; yuxi.zhu@uhhospitals.org; 3Division of Pharmaceutics and Pharmacology, College of Pharmacy, The Ohio State University, Columbus, OH 43210, USA; 4Department of Biomedical Informatics, College of Medicine, The Ohio State University, Columbus, OH 43210, USA

**Keywords:** lipid nanoparticles (LNPs), CRISPR, solid tumors, genome editing, translational research

## Abstract

Solid tumors, with their intricate cellular architecture and genetic heterogeneity, have long posed therapeutic challenges. The advent of the CRISPR genome editing system offers a promising, precise genetic intervention. However, the journey from bench to bedside is fraught with hurdles, chief among them being the efficient delivery of CRISPR components to tumor cells. Lipid nanoparticles (LNPs) have emerged as a potential solution. This biocompatible nanomaterial can encapsulate the CRISPR/Cas9 system, ensuring targeted delivery while mitigating off-target effects. Pre-clinical investigations underscore the efficacy of LNP-mediated CRISPR delivery, with marked disruption of oncogenic pathways and subsequent tumor regression. Overall, CRISPR/Cas9 technology, when combined with LNPs, presents a groundbreaking approach to cancer therapy, offering precision, efficacy, and potential solutions to current limitations. While further research and clinical testing are required, the future of personalized cancer treatment based on CRISPR/Cas9 holds immense promise.

## 1. Introduction

In recent years, the integration of clustered regularly interspaced short palindromic repeats (CRISPR) technology with lipid nanoparticle (LNP) delivery systems emerged as a novel anti-cancer strategy. This innovative approach combines the precision of the CRISPR system for gene editing with the efficacy of LNPs in targeting solid tumors, offering a promising avenue for personalized cancer treatment. 

The CRISPR system, originally a powerful gene-editing tool, has found wide-ranging applications in cellular gene editing, gene expression regulation, and the creation of gene knockout animal models [1]. Furthermore, it has been used in cancer therapeutics by enabling the editing of multiple genes, thereby offering innovative solutions for inhibiting tumor growth and enhancing our understanding of cancer mechanisms. As we delve deeper into the intricacies of tumor biology, the role of the CRISPR system in unraveling the genetic underpinnings of various cancers becomes increasingly apparent [2]. This technology has begun to be applied in the treatment of various types of cancer. In 2016, the first clinical trial involving CRISPR/Cas9 was conducted, demonstrating its tremendous potential in the treatment of genetic disorders and cancer [3]. More recently, the UK approved the first CRISPR-Cas9 gene editing therapy for the treatment of sickle cell disease in 2023. This landmark approval marked a significant advancement in the application of gene editing technologies for therapeutic purposes.

However, the full potential of CRISPR technology is hindered by certain limitations, including off-target effects, sequence restrictions, and low delivery efficiency. These challenges necessitate a cautious approach and rigorous evaluation before the application of gene-editing tools in the human body. To harness the benefits of CRISPR technology, one of the pivotal aspects lies in the efficient and precise delivery of the gene-editing components to target cells and tissues [4,5]. In this context, nanomaterials, particularly LNPs, have emerged as a promising choice as carriers for the CRISPR/Cas9 system [6,7]. LNPs offer tunability, biocompatibility, and the ability to enhance drug delivery efficiency. Researchers have developed innovative LNP formulations to efficiently transport CRISPR components, such as Cas9 plasmids, mRNA, or RNP, to tumor tissues. Nevertheless, the delivery of nucleic acid drugs, including the CRISPR system, still presents challenges, particularly when targeting extrahepatic tissues. Recent advancements in designing nanoparticles that effectively target non-hepatic tissues have shown promise, but the design of delivery carriers must prioritize simplicity for clinical translation.

This review delves into the emerging role of CRISPR technology combined with LNPs in the realm of oncologic research, particularly focusing on pre-clinical studies. It underscores the potential of LNP-CRISPR in reshaping the landscape of cancer research by offering new avenues for gene-level interventions. By exploring recent breakthroughs and advancements, the article emphasizes the significance of LNP-CRISPR technologies in tackling the complexities of cancer treatment and underscores their capacity to lead the way toward more targeted, personalized therapies in the battle against cancer. 

## 2. The Potential of CRISPR-Based Therapy in Cancer Treatment

### 2.1. Structure and Feature of CRISPR System

The CRISPR/Cas9 system, discovered as a natural mechanism employed by archaea and bacteria for their adaptive immune response, functions to protect against external genetic elements such as plasmids and bacteriophages [1]. The mechanisms of CRISPR/Cas9 system are shown in Figure 1. It operates by generating precursor CRISPR RNA (pre-crRNA), which, with the help of trans-activating CRISPR RNA (tracrRNA), forms a complex with Cas9 protein to target and cleave DNA sequences adjacent to the protospacer-adjacent motif (PAM). Although the advent of sgRNA has streamlined this process, delivering the CRISPR components to target cells is crucial for effective gene editing, which can be largely influenced by the forms of CRISPR systems [8,9].

In CRISPR applications, the delivery of the CRISPR system into target cells is achieved through three primary methods: (1) plasmids delivering Cas9 DNA and sgRNA: this method is straightforward and stable but carries a risk of integrating the DNA into the genome, which could lead to unintended genetic alterations [10]; (2) Cas9 mRNA and sgRNA delivery: this approach provides a transient expression of Cas9, reducing the duration of its activity to limit off-target effects but at the expense of stability. The temporary nature of mRNA expression necessitates precise timing and control over the editing process, making it suitable for applications where short-term expression is desired [11]; and (3) ribonucleoprotein (RNP) delivery of Cas9 and sgRNA: delivering the Cas9 protein directly with sgRNA as an RNP complex offers immediate gene editing capabilities with reduced off-target risks. This method circumvents the need for Cas9 transcription and translation within the target cell, leading to a faster onset of editing. However, the purification of RNP complexes and the potential for immune reactions pose significant challenges. Despite these hurdles, RNP delivery is often preferred for its rapid action and precision, regardless of the level of Cas9 in the target cells, making it an attractive option for therapeutic applications requiring immediate correction of genetic errors [12,13,14]. Each method balances efficiency and safety in gene editing, reflecting a nuanced approach in CRISPR technology applications.

### 2.2. CRISPR-Based Technologies for Cancer Treatment

The etiology of tumors is often linked to unregulated cell growth mechanisms, notably through the activation of oncogenes and the deactivation of tumor-suppressor genes. Genome engineering, particularly through CRISPR/Cas9 technology, has emerged as a beacon of hope in cancer therapeutics, given its ability to edit multiple genes. Utilizing CRISPR/Cas9 to deactivate oncogenes has shown efficacy in impeding tumor growth. Conversely, restoring the functionality of tumor suppressor genes can also impede tumorigenesis. Currently, CRISPR/Cas9-based gene therapies are being actively explored in various cancers including lung, breast, colorectal, and hepatocellular carcinoma.

Research on the role of genes in the development of tumors has been greatly enhanced by the use of CRISPR/Cas9 technology. This revolutionary tool has been instrumental in creating cell and animal models with specific gene mutations, which has significantly improved our understanding of disease mechanisms and potential therapeutic strategies. For example, CRISPR/Cas9 has been utilized to unravel the pathogenesis of diseases and identify the roles of new oncogenes or tumor suppressor genes. Specifically, CRISPR/Cas9 was instrumental in uncovering the role of *MLL3*, located on the long arm of chromosome 7, as a tumor suppressor gene within the context of leukemia [13,14,15,16]. In addition to hematologic malignancies, CRISPR/Cas9 technology has also been applied to the study of solid tumors. For example, Dr. Zhang Feng’s team introduced a method for creating mouse models with gene knockouts in tumors using CRISPR/Cas9 [17]. They injected Cas9 and sgRNA-containing DNA plasmids into the livers of wild-type mice via intravenous injection. Editing of the *Pten* and *P53* tumor suppressor genes, either individually or simultaneously, using CRISPR/Cas9 led to increased Akt phosphorylation levels and liver cell lipid accumulation. They replicated this liver cancer model phenotype through traditional gene editing with the Cre-loxP system. DNA sequencing of mouse liver and tumor tissues revealed mutations in tumor cells, including *Pten* and *p53* tumor suppressor gene insertion or deletion of both alleles. Additionally, using CRISPR, they induced β-catenin mutations, leading to liver carcinogenesis. Their study showcases the potential of CRISPR/Cas9 for direct oncogene or tumor suppressor gene editing in mouse liver cells, offering a novel approach to creating liver cancer models and advancing genomic research. Following the successful construction of a mouse liver cancer model using the CRISPR/Cas9 system, they edited multiple genes in various cell types to create a lung adenocarcinoma mouse model [18]. They also used the Cre-loxP system to knock in the Cas9 gene for in vivo gene editing. Furthermore, multiple studies have utilized the CRISPR/Cas9 system to investigate the mechanisms of solid tumors, which may provide novel therapeutic strategies. For example, researchers used CRISPR/Cas9 to knock out the *PIK3R1* gene in rectal cancer cell lines. This resulted in changes related to epithelial-to-mesenchymal transition (EMT), cell proliferation, and tumor cell stemness, highlighting the regulatory role of the *PIK3R1* gene in the invasive, metastatic properties, and tumor stem cell characteristics of rectal cancer cells. To date, an increasing number of tumor-related mutations are being revealed through CRISPR technology. Extensive progress has been made in pancreatic carcinoma [19], cervical cancer [20], osteosarcoma [21], and melanoma [22]. Furthermore, the advent of innovative methodologies such as PERTURB-seq, which combines CRISPR-based genetic screening with single-cell RNA sequencing, has markedly enhanced the efficiency of phenotyping, offering deeper insights into the genetic underpinnings of cancer and paving the way for more precise therapeutic interventions [23]. 

In addition to single gene targeting, CRISPR/Cas9 facilitates large-scale cancer gene screening, enhancing the efficiency of anticancer drug development [24,25,26]. For example, genome-scale CRISPR screening was used to construct a cancer-dependent gene resource library, systematically prioritizing new targets for various tissues and genotypes [24]. An extensive sgRNA library targeting 18,080 genes in the human genome has been developed and named after the GecKo (Genome-scale CRISPR-Cas9 Knockout) Library. Within this library, designers created three to four sgRNAs for each gene, resulting in a total of 64,751 distinct sgRNAs. By combining lentiviral transduction and high-throughput sequencing methods, they identified essential genes associated with melanoma drug resistance and lung cancer metastasis.

### 2.3. Challenges Associated with CRISPR-Based Technologies 

As mentioned above, the CRISPR gene editing system has gradually evolved into important tools for genome editing. However, the CRISPR/Cas9 system faces some challenges that are similar to those of other gene therapies. 

Off-target effects remain a paramount concern, as the specificity of sgRNAs (targeting merely 20 base pairs) leaves room for unintended edits within the vast human genome. Addressing this issue necessitates refined strategies to ensure accuracy without compromising efficacy. Techniques such as employing paired Cas9 nickases, which require dual sgRNA recognition to initiate a double-strand break, thus reducing off-target risks, and enhancing Cas9 fidelity through protein engineering are at the forefront of these efforts. Additionally, meticulous sgRNA design and target site selection is critical for minimizing off-target interactions, underscoring the need for a balanced approach between editing efficiency and improving specificity. The strategies of how to minimize the off-targeting efficiency has been extensively reviewed elsewhere [27].

The lack of a safe and efficient delivery system is another major obstacle to using CRISPR/Cas9 technology for targeted in vivo disease treatment. As a critical CRISPR system effector, functional pDNA had a very short half-life after systemic administration [28]. Several clinical trials showed that naked DNA is not able to induce the desired efficiency, largely due to its pharmacokinetic profile [29]. The rapid and extensive accumulation of pDNA in non-parenchymal cells in the liver was observed, primarily due to serum nuclease activity. Additionally, the activation of the innate immune response posed another challenge [30]. While mRNA could partially evade this response through chemical modifications, pDNA, derived from biological sources, was less amenable to such alterations [31]. Despite extensive chemical modifications aimed at overcoming serum nuclease activity and immune stimulation, a significant challenge persisted: renal filtration, resulting in the excretion of over 50% of oligonucleotide doses in urine, significantly compromising the bioavailability of nucleic acids. Additionally, nucleic acids’ physicochemical characteristics, such as their substantial size and negative charge density, posed a further impediment by hindering diffusion across the cell plasma membrane. The process of internalization via endocytosis presented another set of issues; it often led to lysosomal fusion, where nucleic acids would degrade, necessitating escape from the endosome to reach the cell cytoplasm for DNA or RNA vectors to become biologically active [32]. For CRISPR therapeutics, entering the nucleus posed an additional barrier, as passive diffusion did not occur unless cell division temporarily compromised the nuclear membrane.

To overcome these challenges, various delivery systems have been explored. Both viral and non-viral vectors have been developed to improve the stability and improve the gene-editing efficiency of the CRISPR system. Among the non-viral options, LNPs have shown promise by enhancing nucleic acid stability and facilitating targeted delivery, thereby improving the efficiency of CRISPR-Cas9 systems. LNPs offer a versatile platform for encapsulating nucleic acids, protecting them from enzymatic degradation, and promoting cellular uptake and endosomal escape. In the following sections, we will focus on how LNP, a novel non-viral nanomaterial, enhances the delivery and improves the effectiveness of CRISPR, fostering advancements in treatment.

## 3. The Clinical Potential of LNPs in Overcoming the Delivery Barriers to Solid Tumors

### 3.1. Development of LNPs

LNPs, initially employing liposomes, emerged in the mid-1960s as carriers for small-molecule drugs, proteins, and nucleic acids. The discovery in 1976 of cationic polymer nanoparticles further propelled their use for nucleic acid delivery. LNPs have since evolved to deliver various nucleic acids, including DNA, siRNA, and mRNA, with ongoing optimization of lipid chemistry and formulations. Notably, in 2018, the FDA-approved “Onpattro” (patisiran) targeted hereditary transthyretin-mediated amyloidosis [33]. Recent milestones include Pfizer and Moderna’s COVID-19 vaccines, both of which utilize LNP technology and gained emergency use authorization in 2020, followed by FDA approval [34,35].

LNPs typically comprise four key components: cationic/ionizable lipids, helper phospholipids, cholesterol, and PEG-lipids (Figure 2). Each of these plays a distinct role in enhancing the stability, transfection efficiency, and safety of LNPs. Cationic/ionizable lipids are crucial as they influence the encapsulation and internalization behavior of LNPs. Initially, cationic lipids were used for their positive charge to complex with nucleic acids. However, this posed challenges such as cytotoxicity and stability. Ionizable lipids, with pH-responsive properties, address these issues. These lipids are characterized by a nitrogen-containing head group capable of becoming positively charged in acidic environments while remaining neutral at physiological pH. This head group is connected to a hydrophobic tail, typically consisting of long-chain fatty acids, through a linker that can be a stable or biodegradable bond such as an ester, ether, or amide linkage. The design of these lipids, including the head group, linker, and hydrophobic tail, is carefully optimized to achieve effective nucleic acid delivery. Notable clinical approval involving ionizable lipids are SM-102 [36] and ALC-0315 [37], which are developed by Moderna and Pfizer-BioNTech, respectively. Another clinically approved ionizable lipid is MC3, which serves as a key component of Onpattro [34]. In preclinical studies, many ionizable lipids, such as C12-200 [38], cKK-E12 [39], TT3 [40], 306Oi10 [41], and CAP2 [42], have been developed with specific functions.

Helper phospholipids, on the other hand, are instrumental in wrapping nucleic acids and are considered to be a backbone of the lipid bilayers. Typical phospholipids include DSPC, DOPE, DOTC, DOTMA, and POPC. Among them, DSPC has been in clinics as a key component in the COVID-19 vaccines [43]. Another helper lipid is cholesterol, which is utilized in LNPs for structural integrity. Interestingly, cholesterol derived lipids have shown improved mRNA delivery and certain tissue targeting, highlighting the adaptability of the geometry of cholesterol [44,45]. Lastly, PEG-modified lipids play a crucial role in regulating cellular uptake and prolonging circulation half-life in vivo. PEG-lipids, such as PEG2000-DMG and PEG2000-DSG, prevent nanoparticle aggregation and adsorption of serum proteins, which is critical for maintaining the stability of LNPs during storage and circulation [46].

Currently, the field of high-throughput screening for LNPs is rapidly advancing to meet the growing demand for drug delivery. This screening process aims to identify the most effective lipid combinations to enhance the stability, delivery efficiency, and biocompatibility of LNPs. For instance, researchers are leveraging combinatorial libraries to systematically generate and test a vast array of lipid formulations [34,47,48,49]. This approach allows them to explore numerous lipid combinations efficiently. More recently, deep learning models have been employed to analyze the extensive datasets generated from these experiments [50]. These models can uncover complex relationships between lipid composition and LNP performance that may not be immediately apparent through traditional analysis methods.

### 3.2. LNP-Based Systems for CRISPR Delivery

The application of LNPs in delivering various forms of the CRISPR/Cas9 system, including pDNA, mRNA, and RNP complexes, is an innovative frontier in gene editing technology (Figure 3). Each form has distinct characteristics and requires specific LNP formulations for effective delivery.

In terms of pDNA delivery, LNPs aim to facilitate the transportation of Cas9 and gRNA plasmids into the nucleus, which presents a significant challenge due to the large size of CRISPR plasmids. Their substantial size can impact encapsulation efficiency and interactions with the cellular membrane, including pX260 and pX330. For instance, Kulkarni et al. improved LNP-pDNA delivery by utilizing a formulation that included DLin-MC3-DMA. Additionally, unsaturated phosphatidylcholine (PC) helper lipids were replaced by the saturated helper lipid DSPC, which may increase the membrane fluidity and improve the release kinetics of the therapeutic payload. This modified formulation demonstrated considerably enhanced transfection efficiency and cell viability when compared to conventional reagents, such as Lipofectamine, especially in primary embryonic mesenchymal cells. Another strategy employed by Liang et al. involved compacting the plasmids encoding Cas9-sgRNA using polymer modified lipids, resulting in a lipopolymer structure. This approach resulted in a 50% reduction in gene knockout efficiency targeting VEGF when comparing the therapeutic group to the control group with no active payload [51]. Another similar study developed PLGA modified lipids for gene knockout on macrophages, achieving gene editing efficiencies of 30% and 20% in vitro and in vivo, respectively [52].

mRNA can be translated into proteins directly in the cytoplasm, eliminating the need for entry into the cell nucleus. When delivering Cas9 mRNA and gRNA, LNPs offer a distinct advantage by leveraging their ability to efficiently cross cell membranes and protect the mRNA from degradation. One of the initial strategies involved the combination of LNP and AAV delivery techniques to facilitate the Casp mRNA and sgRNA, respectively. This method capitalizes on the LNPs’ capacity for efficient and transient mRNA delivery, ensuring temporary expression of Cas9 to minimize potential off-target effects. Concurrently, AAV vectors are utilized for their proven efficacy in targeting a wide range of tissues and their ability to carry the necessary components for precise gene editing. This approach achieved gene editing in approximately 6% of hepatocytes, demonstrating the effectiveness of this combined delivery in genome editing [47]. More recently, co-encapsulating Cas9 mRNA and gRNA in LNPs represents another strategy to enhance genome editing effectiveness. Liu et al. demonstrated this using an LNP formulation containing a bioreducible ionizable lipid (BAMEA-O16B), achieving a 90% reduction in cellular GFP expression in vitro [53]. In vivo, this formulation significantly reduced serum PCSK9 levels in mice without showing signs of inflammation or hepatocellular injury. Similarly, Finn et al. showcased the capability of LNPs to effectively deliver both mRNA and gRNA in vivo, leading to long-term gene editing outcomes [54]. Furthermore, Jiang et al. harnessed N1,N3,N5-tris(2-aminoethyl)benzene-1,3,5 tricarboxamide (TT3)-derived lipid-like nanoparticles (LLNs) for the delivery of Cas9 mRNA and sgRNA, demonstrating their therapeutic significance by effectively disrupting *HBV* genes in mouse models [55]. In another study, Miller et al. developed a series of zwitterionic amino lipids and demonstrated their ability of safe and long-term delivery of Cas9 mRNA and sgRNAs [56].

Delivery of Cas9/gRNA RNP complexes via LNPs represents another promising strategy. This approach is believed to offer superior editing efficiency and target specificity with fewer off-target effects. For example, Wang et al. employed an LNP formulation containing bioreducible lipids to deliver RNPs targeting GFP-expressing HEK cells. This formulation facilitated the escape of the LNP cargo from endosomes and resulted in efficient genome editing with minimal off-target effects. In summary, LNP-mediated delivery of CRISPR/Cas9 in its pDNA, mRNA, and RNP forms has shown remarkable potential. Examples, such the work of Kulkarni et al., Zhang et al., and Liu et al., illustrate the advancements in this field, each tailored to overcome the specific challenges associated with the different forms of CRISPR/Cas9 components. These innovations pave the way for more efficient and precise genome editing applications.

### 3.3. Application of LNP-CRISPR Systems for Cancer Treatment

In recent years, the convergence of CRISPR technology with LNP delivery systems has marked a significant milestone in the realm of cancer therapy. This innovative approach leverages the precision of CRISPR for gene editing and the efficacy of LNPs in targeting multiple solid tumors including brain tumors, melanoma, liver, and ovarian cancers (Figure 4). This section provides an overview of the key examples where LNP-mediated CRISPR delivery has been employed in tumor therapy.

Brain Tumor Therapeutics: solid tumors such as glioblastoma present unique challenges due to the blood-brain barrier (BBB), a formidable obstacle to drug delivery. The integration of LNP technology with CRISPR gene editing has shown remarkable progress in brain tumor therapeutics. For example, the development of lipid-coated mesoporous silica nanoparticles (LC-MSNs) has been developed for enhanced CRISPR delivery. Notably, in the in vivo study, these LC-MSNs were employed for intrastriatal injections in an Ai9-tdTomato reporter mouse model. Remarkably, this approach achieved a gene editing efficiency of approximately 10 ± 2% in the targeted brain tissue in a tdTomato mouse model, demonstrating the nanoparticles’ potential in delivering CRISPR components in vivo [57]. Another significant study in brain tumor therapy involved the utilization of LNPs to deliver mRNA encoding Cas9 protein directly into the brain [58]. Notably, the study achieved significant gene editing in both the striatum and hippocampus, demonstrating a high transfection efficiency. High doses of MC3 LNP Cas9 mRNA/Ai9 sgRNA resulted in substantial gene editing, with a remarkable increase in total tdTomato+; DAPI+ cells, showcasing the potential of this approach in treating brain tumors. This approach, while promising, raises critical considerations regarding the LNP’s ability to achieve sustained gene editing in the context of the brain’s delicate and complex microenvironment, especially through systemic administration. The incorporation of targeting moieties, or the utilization of neurotransmitter derived LNPs, presents a promising strategy for overcoming the challenges of brain delivery. However, their effectiveness in CRISPR delivery requires further investigation to be confirmed.

Melanoma Treatment: melanoma, characterized by its aggressive nature and tendency for early metastasis, poses significant challenges for therapeutic delivery. The exploration of CRISPR-LNP systems has extended to melanoma treatment. For example, a novel LNP system encapsulating CRISPR/Cas9 was designed for targeting the *PLK-1* gene in melanoma cells [59]. This core-shell nanoparticle structure facilitated efficient gene editing, showing remarkable in vivo results. In melanoma-bearing mice, the PLNP/DNA-a treatment significantly reduced tumor growth, achieving about 67% tumor volume inhibition compared to the PBS group and 50% compared to the PLNP/siPLK-1 group. Deep sequencing indicated a wide range of indels (1–28 bp) in the tumor genomic DNA. These results underscore the efficacy of the LNP-CRISPR system, marking a substantial advance in melanoma therapy. In order to further achieve the potential of CRISPR/Cas9 gene editing, combination therapy has been used to maximize the anti-tumor efficiency. For example, research introduces a unique lipid-encapsulated gold nanoparticle (AuNPs) system for delivering the CRISPR/Cas9 [60]. This system, designed to target the *Plk-1* gene in melanoma cells, utilizes AuNPs for photothermal release of the CRISPR/Cas9 components. In vivo results showed that the treatment significantly reduced tumor growth in mice, with the tumor volume reducing to about 15% of the control group after laser irradiation. This study not only demonstrates a novel approach in melanoma therapy but also showcases the potential of combining photothermal therapy with the CRISPR-LNP system. Currently, due to melanoma’s adeptness at evading the immune system, there is a growing consensus that combining CRISPR-based interventions with immunotherapy could offer a more effective strategy. However, related research is still limited and the application of targeted LNPs for delivering CRISPR tools in such a combined therapy approach requires thorough investigation.

Liver and Ovarian Cancer Therapy: further advancements in CRISPR-LNP technology were demonstrated in the treatment of liver and ovarian cancers. For example, researchers developed a novel ionizable lipid nanoparticle (iLNP) called iLP181 for delivering the CRISPR/Cas9 system, specifically targeting the PLK1 gene in liver cancer [61]. They constructed a plasmid, named psgPLK1, which expresses both the Cas9 protein and the sgRNA. The iLP181/psgPLK1 nanoformulation exhibited uniform size, near-neutral zeta potential at physiological pH, and effective endosomal escape, ensuring efficient delivery and gene editing. In vivo, the treatment led to significant tumor growth suppression in a mouse model of liver cancer, without inducing adverse effects on major organs. This study highlights the potential of iLP181 LNPs as a clinically applicable system for delivering CRISPR/Cas9 and treating liver cancer effectively. In another study focused on ovarian tumors, researchers employed a targeted nonviral LNP system for therapeutic genome editing using CRISPR-Cas9 [62]. They developed LNPs encapsulating Cas9 mRNA and sgRNA using a novel ionizable amino lipid. This system achieved high gene editing efficiency in vitro and in vivo. Specifically, in an ovarian tumor model, the targeted LNPs exhibited up to approximately 80% gene editing efficiency in tumor cells, significantly inhibiting tumor growth and improving survival. This research highlights the potential of targeted CRISPR-Cas9 LNPs for treating metastatic cancers such as ovarian tumors. Relative to brain tumors or melanoma, other solid tumor delivery, especially liver-targeted delivery, presents some advantages for LNPs, primarily due to the liver’s natural propensity to accumulate nanoparticulate systems. This unique feature can facilitate the more efficient delivery of CRISPR components to liver tissues. However, the challenge remains to ensure specificity and minimize off-target effects, particularly in avoiding unintended gene editing in non-cancerous liver cells.

## 4. Conclusions and Perspectives

The CRISPR/Cas9 technology, a powerful gene editing tool, is widely applied in various fields such as cellular gene editing, gene expression regulation, construction of gene knockout animal models, and therapeutic research for cancer treatment. With the establishment of high-throughput library screening techniques, CRISPR/Cas9 technology has provided promising solutions for understanding the mechanisms of tumor development, drug discovery, and cancer treatment. Despite the numerous advantages of CRISPR/Cas9 technology, it still faces limitations, such as off-target effects and low transduction efficiency, which hinder its full potential. Therefore, the application of CRISPR/Cas9 technology should focus on establishing effective in vivo delivery systems.

Nanomaterials, such as LNPs, have shown promise as carriers for the CRISPR/Cas9 system. One of the primary advantages of LNPs is their ability to encapsulate nucleic acids, protecting them from enzymatic degradation and facilitating targeted delivery into cells. This has been critical in the development of mRNA vaccines for COVID-19, where LNPs enable the safe and effective delivery of the genetic material into muscle cells. Moreover, LNPs provide a versatile and tunable platform which can realize multiple therapeutic purposes. LNP components can be designed with different chemical structures to achieve specific functionalities, such as activating the immune system or aiding in therapeutic delivery. By tailoring the composition of the lipid bilayer, including the selection of ionizable lipids, helper lipids, cholesterol, and PEGylation, researchers can manipulate the physicochemical properties of LNPs, such as their size, charge, and encapsulation efficiency. Additionally, the incorporation of targeting ligands on the surface of LNPs can further enhance their specificity, enabling precision medicine approaches for various diseases.

However, LNP still faces some challenges when delivering nucleic acid drugs, including via the CRISPR system. Firstly, delivering gene-editing components to extrahepatic tissues remains challenging. Recent efforts in designing nanoparticles targeting non-hepatic tissues, including tumor tissues, have been increasingly successful. In some cases, non-targeted nanoparticles without ligands may demonstrate effectiveness when administered locally through intratumorally injection, spinal injection, or inhalation delivery. However, achieving clinical-grade dosage administration remains a challenge. Moreover, it is important to note that certain tumor cell types may necessitate active targeting strategies for enhanced therapeutic outcomes. For instance, tumor cells characterized by the high expression of tumor-specific antigen (TSA) and tumor-associated antigens (TAAs) may benefit the specificity and uptake of LNPs modified with ligands targeting on these antigens. Examples include targeting HER2 for breast cancer [63] and utilizing folate receptors [62] for the treatment of certain types of ovarian cancer. However, when considering the addition of active targeting ligands, it is essential to assess whether the complexity added for improved selectivity or specificity is justified. To avoid unnecessary surprises in later stages of preclinical research, optimized targeted and non-targeted administrations in mice must be confirmed in corresponding studies in non-human primates. Despite much data from mice, and an increasing amount from non-human primates, suggesting that clinically relevant delivery to the lungs and the immune system is feasible after intravenous administration, extensive in vivo delivery characterization, including local and systemic toxicity, needs to be rigorously evaluated. 

The clinical applications of LNPs have expanded rapidly, most notably with the FDA’s approval of mRNA vaccines for COVID-19, which has paved the way for LNPs in mainstream medicine. Beyond vaccines, LNPs are being explored for gene therapy, cancer treatment, and personalized medicine, offering new potential for diseases that were previously difficult to treat. The use of LNPs in delivering CRISPR-Cas9 components for gene editing presents another exciting avenue, potentially enabling the correction of genetic disorders at their source. Looking forward, the application of LNP-CRISPR in the field of solid tumors represents a breakthrough. The continuous optimization and improvement of CRISPR/Cas9 demonstrate its tremendous research and application potential in the future of medicine, including medical research, clinical treatment, and gene drug development. However, most of the current research is still in its infancy, and there are many molecular-level issues to be resolved in the application of CRISPR/Cas9 to cancer treatment. The limited safety and efficacy profiles observed when LNPs are administered systemically in vivo highlight a significant barrier to their widespread clinical adoption. This is compounded by the lack of comprehensive human data, which are crucial for validating the promising results seen in preclinical models. As we move forward, this technology must also undergo rigorous clinical testing, including efficacy, safety, and specificity testing, before it can be applied in clinical practice. Therefore, the journey from preclinical success to clinical application requires not only innovation in delivery mechanisms but also a deep understanding of the interaction between these novel systems and the human body. 

## Figures and Tables

**Figure 1 cells-13-00568-f001:**
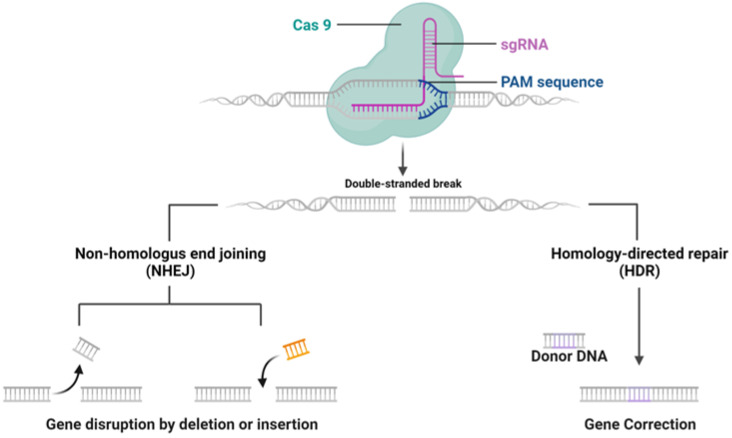
CRISPR-Cas9 mechanism of Action. CRISPR-Cas9 system targeting a DNA sequence adjacent to a PAM sequence, causing a double-stranded break. Two DNA repair pathways are presented: non-homologous end joining (NHEJ) can result in gene disruption by random insertion or deletion, while homology-directed repair (HDR) allows for precise gene correction using a donor DNA template. Figure created by Biorender.

**Figure 2 cells-13-00568-f002:**
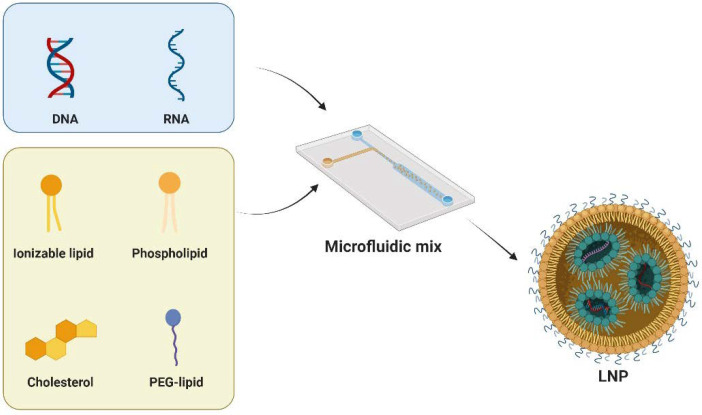
LNPs typically consist of cationic/ionizable lipids, helper phospholipids, cholesterol, and PEG-lipids. Microfluidic techniques are used to precisely control the preparation of LNPs. This involves manipulating fluids at a microscale within small channels or devices. The method enables the controlled mixing of lipid components and the encapsulation of nucleic acids. Figure created by Biorender.

**Figure 3 cells-13-00568-f003:**
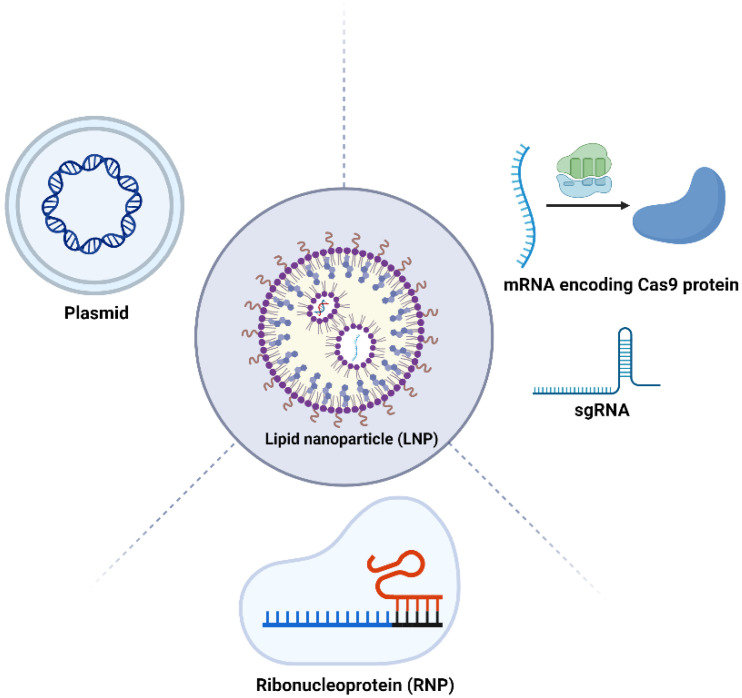
Schematic representation of CRISPR-LNP therapeutics. LNPs can be used for the delivery of plasmids, mRNA delivery, mRNA, sgRNA, and the RNP complex. Figure created by Biorender.

**Figure 4 cells-13-00568-f004:**
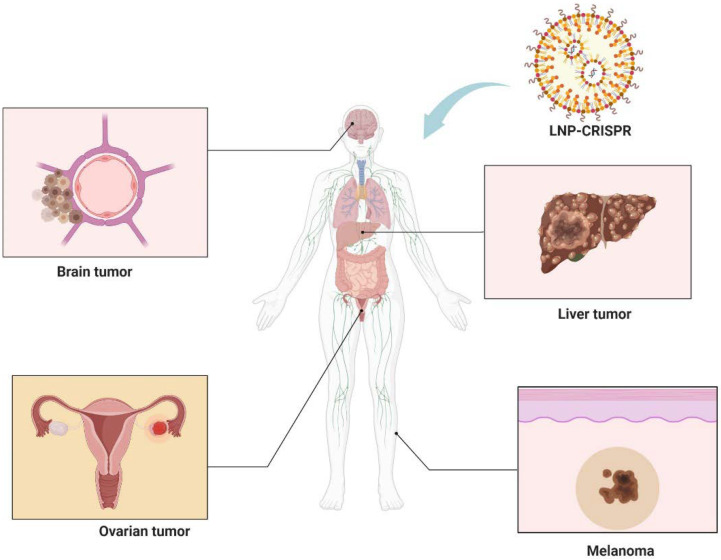
Integration of LNP technology with CRISPR gene editing shows promise in treating brain tumors, melanoma, liver, and ovarian cancers. Figure created by Biorender.

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
