# Peer review of "Preclinical Advances in LNP-CRISPR Therapeutics for Solid Tumor Treatment"

_cells, 2024, doi:10.3390/cells13070568_

Round 1
Reviewer 1 Report
Comments and Suggestions for Authors
CRISPR therapeutics represents a ground-breaking breakthrough and hold tremendous potential in cancer treatment, however, this technology still in its infancy and there are many issues to be resolved in the application of it to clinic. One of the big challenges is to deliver CRISPR to target sites. This review written by Wang et al has summarized recent advance in CRISPR/Cas delivery by LNP, highlighted the superior advantages of LNP in improving delivery efficacy, editing efficiency and target specificity. This review is a prompt review and would provide an useful introduction and guidance for researcher who works and will work on this area.
I recommend acceptance after a few points.
Major points:
This review is largely depleted of figures. For example, a figure can be created for section 3.1 to present the main idea: the development of LNP, the key components of LNP etc. Same suggestion for 3.3. In addition, as the main focus of the review is the LNP used for CRISPR delivery, the history of LNP development, LNP types( possibly the chemical structures), the advantages and disadvantages of each type of LNP, the clinical application and perspectives should be expanded.
Minor points:
Page 1 lane 38-42: This is not up to date. the UK has approved the first CRISPR-Cas9 gene editing therapy for the treatment of sickcle cell disease in 2023.
Page 2 lane 82-84: how cas9 mRNA and sgRNA delivery reduces off-target effect?
Page 4 lane 123-124: typo “creating” “advancing”
Page 4 lane 137-138: references are not up to date. for instance, PERTURB-seq has combined CRISPR-based genetic screening with single-cell RNA-seq, largely improved the phenotyping efficiency. I suggest the authors to update all recent breakthroughs into the review.
Page 4 lane 140-142: lack references
Page 6 lane 247-250: explain the principle of the improvement by changing the saturated helper lipid DSPC with unsaturated PC helper lipids.
Page 7 lane 254-255: by comparing to what 50% gene knockout efficiency is an improvement.
Reviewer 2 Report
Comments and Suggestions for Authors
SUMMARY:
Despite the vast potential oncologic applications of CRISPR technology, there are barriers to delivery. Lipid nanoparticles (LNPs) provide a tunable vehicle for CRISPR systems in-vitro, in-vivo, and potentially in humans. In this review, the authors detail early use of CRISPR technology in cancer research, the evolution of LNPs to overcome barriers to CRISPR delivery, pre-clinical therapeutic applications of LNP-CRISPR systems, and perspectives on the future direction of these systems.
Overall, the paper provides a solid background on the joint use of LNPs and CRISPR technology in oncology research. Current and relevant literature in the field was adequately cited. However, the paper failed to accomplish what its title suggested. Most of the review focused on the development of each technology while only scratching the surface on therapeutic applications. Furthermore, the paper superficially explored solid tumor applications. The paper accurately summarizes the current state of LNP technology in its conclusion: more research into safety and efficacy in humans is required. Though the general structure is in place, this review serves more as a general summary than as an insightful review.
Suggestions for improvement revolve around organization, scope, and language. First, the authors must consider who their audience is. Most readers looking for a literature review that specifically addresses LNPs as delivery system for CRISPR/Cas9 for solid tumor treatment likely already have a general understanding of CRISPR and LNP technologies. Thus, a significant amount of elementary background can be removed. Instead, more information regarding current delivery systems i.e. electroporation and its limitations can be described in the CRISPR section. By limiting background, greater emphasis can be placed on describing pre-clinical CRISPR literature, nuances in delivery limitations based on each CRISPR system (plasmid DNA versus mRNA versus RNP) and literature involving each, and the nuances each LNP-CRISPR project had to overcome specific barriers. Secondly, the authors must decide which application to focus on: pre-clinical models studying a variety of oncologic disorders, potential therapeutic applications, or research into solid tumors. All three applications have various details that were only superficially explored in this review. This was most notable with regards to solid tumors. The preclinical literature into brain tumor, melanoma, liver, and ovarian cancers served as the heart of the paper yet composed only a minority. By rearranging the scope or organization, more time can be dedicated to the cited literature applications. Lastly, a frequent use of flowery descriptive language took away from the overall execution of the paper. The use of LNPs to augment CRISPR technology is an evolving field. With improvements, this paper can serve as a concise literature review that contributes significantly to the field.
MAJOR COMMENTS:
1. The paper does not thoroughly review LNP-CRISPR therapeutics for solid tumor treatment. The scope should be changed to use of LNP-CRISPR technologies in pre-clinical oncologic research, or more focus should be given to solid tumor literature.
2. Stating “solid tumor treatment” in the title implies there is translational research. If this remains the focus, the title should be reworded to acknowledge that literature remains preclinical.
3. The introduction, CRISPR, and LNP backgrounds should be shortened based on the target audience. Instead, more time can be dedicated to defining the problems in CRISPR-Cas9 delivery systems as this is the focus of the paper.
4. More time is required reviewing the nuances between pre-clinical literature that was cited in the “application of LNP-CRISPR system for cancer treatment” section. This includes assessing differences in the CRISPR systems and LNPs used, rationale for modifications, and critical appraisal of their results.
5. Similarly, more time can be dedicated in the future perspectives sections on the literature cited, specifically detailing the rationale for each project in relation to LNP barriers i.e. limited safety and efficacy profile when administered systemically in-vivo, lack of human data.
MINOR COMMENTS:
1. Clarify line 65-66: “integral to the adaptive immune response in archaea and bacteria” can be re-written to acknowledge CRISPR technology was discovered from natural processes in archaea/bacteria.
2. Clarify why the example of MLL3 and leukemia is used in line 110. Understanding that its discovery was related to CRISPR technology was not intuitive and required this reviewer to read the cited paper.
3. Clarify how GFP knockout by CRISPR is a clinically relevant application in line 264.
4. Repetitive language in line 127-128; a similar statement regarding use of CRISPR in investigating solid tumors precedes this in lines 111-112.
5. Clarify line 258-259: it is not LNPs, but mRNA that has an advantage over DNA by not requiring nuclear entry for translation.
6. Line 260: elaborate the combination of LNP and AAV vehicles in the text. This reviewer had to refer to the paper to understand why it was cited. The paper revealed that the LNP carried cas9 mRNA while the AAV encoded sgRNA and repair templates. The review also does not describe the initial rationale for separating vectors and how the following citation used LNPs to carry both cas9 and guide RNA.
7. Lines 295-301: clarify how CRISPR served a therapeutic role in the mouse model.
8. Line 136: remove “etc.”
9. Line 145: make “essentials” singular.
10. Line 149 “one of the most powerful,” lines 378 “groundbreaking, ” line 321 “hope” and “impressive” are examples of overly descriptive language that is not required in this review.
11. Line 164, line 215, and line 321: improper use of the word “besides.”
12. Line 270 and 273: change the words “further” and “furthermore” or sentence syntax to avoid repetition.
Comments on the Quality of English Language
Some minor editing of English language is required throughout the manuscript.
